# Efficacy of HDAC Inhibitors in Driving Peroxisomal β-Oxidation and Immune Responses in Human Macrophages: Implications for Neuroinflammatory Disorders

**DOI:** 10.3390/biom13121696

**Published:** 2023-11-23

**Authors:** Andrea Villoria-González, Bettina Zierfuss, Patricia Parzer, Elisabeth Heuböck, Violetta Zujovic, Petra Waidhofer-Söllner, Markus Ponleitner, Paulus Rommer, Jens Göpfert, Sonja Forss-Petter, Johannes Berger, Isabelle Weinhofer

**Affiliations:** 1Department of Pathobiology of the Nervous System, Center for Brain Research, Medical University of Vienna, 1090 Vienna, Austria; andrea.villoriagonzalez@meduniwien.ac.at (A.V.-G.);; 2Department of Neuroscience, Centre de Recherche du CHUM, Université de Montréal, Montréal, QC H2X 0A9, Canada; 3Institut du Cerveau—Paris Brain Institute—ICM, Inserm, CNRS, APHP, Hôpital Pitié Salpétrière—University Hospital, Sorbonne University, DMU Neuroscience 6, 75013 Paris, France; 4Division of Immune Receptors and T Cell Activation, Institute of Immunology Center for Pathophysiology, Infectiology and Immunology, Medical University of Vienna, 1090 Vienna, Austria; 5Department of Neurology, Comprehensive Center for Clinical Neurosciences and Mental Health, Medical University of Vienna, 1090 Vienna, Austria; 6Department of Pharma and Biotech, NMI Natural and Medical Sciences Institute, University of Tübingen, 72770 Reutlingen, Germany

**Keywords:** ABCD1, entinostat, foam cell, peroxisome, multiple sclerosis, tefinostat, very long-chain fatty acids, vorinostat, X-linked adrenoleukodystrophy

## Abstract

Elevated levels of saturated very long-chain fatty acids (VLCFAs) in cell membranes and secreted lipoparticles have been associated with neurotoxicity and, therefore, require tight regulation. Excessive VLCFAs are imported into peroxisomes for degradation by β-oxidation. Impaired VLCFA catabolism due to primary or secondary peroxisomal alterations is featured in neurodegenerative and neuroinflammatory disorders such as X-linked adrenoleukodystrophy and multiple sclerosis (MS). Here, we identified that healthy human macrophages upregulate the peroxisomal genes involved in β-oxidation during myelin phagocytosis and pro-inflammatory activation, and that this response is impaired in peripheral macrophages and phagocytes in brain white matter lesions in MS patients. The pharmacological targeting of VLCFA metabolism and peroxisomes in innate immune cells could be favorable in the context of neuroinflammation and neurodegeneration. We previously identified the epigenetic histone deacetylase (HDAC) inhibitors entinostat and vorinostat to enhance VLCFA degradation and pro-regenerative macrophage polarization. However, adverse side effects currently limit their use in chronic neuroinflammation. Here, we focused on tefinostat, a monocyte/macrophage-selective HDAC inhibitor that has shown reduced toxicity in clinical trials. By using a gene expression analysis, peroxisomal β-oxidation assay, and live imaging of primary human macrophages, we assessed the efficacy of tefinostat in modulating VLCFA metabolism, phagocytosis, chemotaxis, and immune function. Our results revealed the significant stimulation of VLCFA degradation with the upregulation of genes involved in peroxisomal β-oxidation and interference with immune cell recruitment; however, tefinostat was less potent than the class I HDAC-selective inhibitor entinostat in promoting a regenerative macrophage phenotype. Further research is needed to fully explore the potential of class I HDAC inhibition and downstream targets in the context of neuroinflammation.

## 1. Introduction

Neuroinflammation refers to a broad range of immune responses that are specific to the central nervous system (CNS) and involves intrinsic brain cells such as microglia and astrocytes, in addition to extrinsic infiltrating peripheral leukocytes. The initial neuroinflammatory processes are targeted to clear and control the eliciting stimulus by the secretion of reactive oxygen species and inflammatory cytokines, followed by the removal of tissue and lipid-enriched myelin debris by professional phagocytes, such as monocyte-derived macrophages and microglial cells. Under homeostatic conditions, myelin phagocytosis triggers anti-inflammatory reprogramming through the upregulation of peroxisome proliferator-activated receptor γ (PPARγ) and liver X receptor (LXR)-mediated reverse cholesterol pathways, thus limiting inflammation and possibly supporting the local remyelination of adjacent oligodendrocytes [1]. While the removal of myelin debris is essential for remyelination and tissue repair, excessive lipid uptake by phagocytes, indicated by a foamy morphology, is thought to disrupt lipid metabolism and to favour a pro-inflammatory phenotype, as observed in various neurodegenerative and neuroinflammatory disorders including X-linked adrenoleukodystrophy (X-ALD) and multiple sclerosis (MS) [2,3]. 

Myelin debris resulting from demyelination is highly enriched in cholesterol and saturated very long-chain fatty acids (VLCFAs, ≥C22). VLCFAs are normally of low abundance and are solely degraded by β-oxidation within peroxisomes [4]. An overload of cells with myelin-derived lipids could promote the production of potentially toxic cholesterol metabolites, reactive oxygen species, and the formation of cytotoxic cholesterol crystals, all of which have been linked to impaired remyelination and neurodegeneration [5,6,7]. It has been proposed that increased cellular VLCFA levels alter membrane fluidity and stiffness, and they have been associated with the initiation of inflammatory cell death [8,9,10]. In macrophages, pro-inflammatory stimulation is associated with high levels of VLCFAs, which are believed to contribute to shape the cellular membrane for pro-inflammatory responses [11]. During the acute activation phase, healthy macrophages rapidly clear VLCFAs by increased peroxisomal β-oxidation, which prevents the pro-inflammatory skewing of cells [11]. Also, microglial cells react to alterations in the VLCFA metabolism by reprogramming the expression of immune response genes, as recently evidenced by the whole transcriptomics analysis of murine BV-2 microglial cells with genetically inactivated peroxisomal β-oxidation [12]. Therefore, in the brain, peroxisomes are crucial not only for shaping and maintaining the lipid content of neural cell membranes and myelin constituents but also for degrading excess VLCFAs upon the ingestion of myelin debris [13]. 

High levels of VLCFAs are present in tissues and body fluids of patients with the peroxisomal disorder X-ALD, which is caused by mutations in the *ABCD1* gene [14,15]. ABCD1 encodes a transporter that imports VLCFAs into peroxisomes for degradation by β-oxidation. In its severest neuroinflammatory presentation, cerebral ALD (CALD), an unknown factor triggers a life-threatening and rapidly progressive destruction of neurons and myelin in the brain of patients [16,17,18]. Whereas the neuropathology of X-ALD is caused by primary peroxisomal alterations due to inherited *ABCD1* mutations, secondary peroxisomal changes have been identified in MS. Studies of postmortem brain tissue from patients with MS have reported a reduced number of PMP70-positive peroxisomes and *PMP70* transcripts in the cortical grey matter accompanied by the elevation of the saturated VLCFA C26:0 [19], as well as a reduced expression of peroxisomal genes in brain white matter [20]. 

Under homeostatic conditions, the blood–brain barrier restricts the movement of peripheral blood mononuclear cells (PBMCs), allowing only activated T cells to access the cerebrospinal fluid (CSF) areas to ensure immune surveillance in the CNS [21]. However, active CNS lesions in patients with CALD and MS are characterized by extensive immune cell infiltration and demyelination [22,23,24,25]. The infiltrates consist of T cells and macrophages, as well as B cells in patients with MS. How these immune cell infiltrates relate to and probably drive neurotoxicity is unclear. When applied at an early stage of disease progression, the replacement of ABCD1-deficient hematopoietic stem cells (HSC) by allogeneic transplantation or gene therapy of autologous HSC halts the inflammatory course of CALD [26,27]. In MS, autologous HSC-transplantation is under consideration as a novel approach to slow down the progression of the disease in patients with resistance to disease-modifying therapies [28]. 

In CALD, monocytes and differentiated macrophages seem to be central in the neuroinflammatory course of the disease, as they are the HSC-derived immune cells most affected by ABCD1 deficiency regarding VLCFA accumulation, and show pro-inflammatory skewing and impaired plasticity both in vitro and within active CNS lesions [2,29]. Moreover, brain autopsies of patients with CALD revealed the accumulation of phagocytes with a foamy morphology in the actively demyelinating lesion area [30]. The recent single-nuclei RNA-sequencing of postmortem brain tissue from patients with MS demonstrated that the chronic active lesion edge comprises activated microglia, infiltrating monocytes/dendritic cells, and monocyte-derived as well as perivascular macrophages [31]. Interestingly, these chronic active lesions (also referred to as mixed active/inactive) are known to expand in time, and a higher lesion load was found to correlate with a faster disease progression [32,33,34]. Therefore, inflamed foamy microglia and macrophages are thought to accelerate disease progression by impaired lipid processing and the associated accumulation of inflammatory lipids, interfering with remyelination [1,33,34]. Accordingly, the disturbance of macrophage metabolism seems to be among the key drivers in neuroinflammatory, demyelinating diseases such as MS and X-ALD and, thus, may offer untapped therapeutic potential in terms of pharmacological intervention. 

Histone deacetylases (HDACs) are a group of enzymes that remove acetyl groups from histones and other nuclear or cytoplasmic proteins, leading to the modulation of gene transcription and translation [35]. With HDACs being crucial regulators of innate immunity [36], pharmacologic compounds known as HDAC inhibitors (HDACis) have been proposed as drug candidates for the treatment of neuroinflammation and other related CNS disorders [37,38]. Among others, the pan-HDACi vorinostat (SAHA) was found to reduce the incidence and severity of the murine experimental autoimmune encephalomyelitis (EAE) model for MS as well as induce peroxisomal activity in ABCD1-deficient fibroblasts and astrocytes [39,40]. Similarly, we previously demonstrated that vorinostat and the HDAC class I-selective HDACi entinostat (MS-275) can stimulate the peroxisomal degradation of VLCFAs and partially rescue the metabolic defects in X-ALD macrophages [40,41]. Due to the promising effects of these compounds in vitro, three CALD patients with an advanced lesion status, and who were thus not eligible for HSC-transplantation or gene therapy, received vorinostat in a compassionate use setting [40]. Two of the patients had to prematurely withdraw from the treatment due to the development of severe thrombocytopenia, which is a common side effect of HDACis. Intriguingly, the analysis of the CSF and serum of the CALD patient who continued the vorinostat treatment for 80 days indicated a temporary improvement of the blood–brain/blood–CSF barrier integrity. However, a brain MRI scan revealed the further progression of the inflammatory demyelination in this already advanced patient. This raised the question of whether a more selective drug, directly targeting macrophages, would minimize side effects and be more beneficial for the treatment of CALD and possibly other neuroinflammatory disorders with innate immune cell activation and recruitment to the brain. The HDACi tefinostat (CHR-2845) is a vorinostat derivative containing an esterase-sensitive motif (ESM), which is cleaved by carboxylesterase 1 (CES1). The acidic product of this reaction (CHR-2847), due to its charge, accumulates in cells from the monocytic lineage, where CES1 is highly expressed. This delivery technology allows for the selective accumulation of the drug within monocytes, differentiated macrophages, and dendritic cells [42]. Of note, clinical studies in patients affected by advanced hematological malignancies demonstrated a good tolerance and lack of dose-limiting toxicity [42,43]. The application of macrophage-targeted HDACis using the ESM technology has not yet been explored in the context of neuroinflammation. 

In the present study, we show that phagocytes upregulate the peroxisomal genes involved in β-oxidation upon myelin phagocytosis and pro-inflammatory stimulation, and that this response is impaired in macrophages and brain white matter lesions in MS patients. Moreover, we compared the macrophage-selective HDACi tefinostat, the pan-HDACi vorinostat, and the class I HDAC-selective HDACi entinostat in terms of their ability to modulate peroxisome-related gene expression and VLCFA metabolism associated with inflammation in human primary macrophages. Our findings reveal that HDACis significantly stimulated the peroxisomal degradation of VLCFAs and phagocytosis of myelin debris and interfered with monocyte chemotaxis. However, entinostat was the most potent among the tested HDACis, in particular in shifting macrophage metabolism towards reverse cholesterol metabolism, which is associated with the resolution of neuroinflammation and remyelination.

## 2. Materials and Methods

### 2.1. Isolation of Human PBMCs, CD14+ Monocytes, CD3+ T Cells, and CD19+ B Cells

PBMCs, CD14+ monocytes, CD3+ T cells, and CD19+ B cells were isolated from human leukoreduction system chambers, purchased from the General Hospital of Vienna, Austria, using Ficoll density-gradient centrifugation (PAN-Biotech; Aidenbach, Germany). We positively selected for CD14+, CD3+, or CD19+ cells using MACS microbeads and the LS column system (Miltenyi Biotec, Bergisch Gladbach, Germany) according to the manufacturer’s instructions. 

### 2.2. Differentiation and Activation of Human Macrophages

CD14+ monocytes were in vitro differentiated to macrophages by culturing the cells for 7 days in an RPMI medium (Sigma Aldrich, St. Louis, MO, USA) containing 1% penicillin/streptomycin, 1% glutamine, 1% Fungizone (all Invitrogen, Carlsbad, CA, USA), and 10% FCS (Gibco Life Technologies/Invitrogen, Waltham, MA, USA,), supplemented with either 50 ng/mL human macrophage-colony stimulating factor (M-CSF, PeproTech, Rocky Hill, NJ, USA) or granulocyte-macrophage colony-stimulating factor (GM-CSF, PeproTech). After differentiation, macrophages were polarized with 25 ng/mL IFNγ (PeproTech) and/or 100 ng/mL lipopolysaccharide (LPS, Sigma Aldrich) for the specified times. For the experiments involving monocyte-derived macrophages from patients with MS, control and MS CD14+ monocytes were differentiated for 3 days in RPMI 1640 containing 10% FBS and 500 U/mL GM-CSF (ImmunoTools, Friesoythe, Germany). Then, macrophages were polarized with 200 U/mL IFNγ (ImmunoTools) and 10 ng/mL LPS (InvivoGen, Vista Sorrento Pkwy, CA, USA) as described by Fransson and colleagues [44]. The HDACis vorinostat (SAHA, Cat.No.10009929, Cayman Chemicals, Ann Arbor, MI, USA), entinostat (MS-275, Cat.No.T6233, Target Mol, Boston, MA, USA), or tefinostat (Cat.No. HY-106409, MedChemExpress, Monmouth Junction, NJ, USA) were dissolved in DMSO and used at a final concentration of 2 µM for up to 48 h. 

### 2.3. Calcein AM Viability Assay

The viability of human macrophages exposed to vorinostat, entinostat, tefinostat, or the DMSO solvent control was assessed using Calcein AM staining. Cells were incubated with 2 µM HDACi or DMSO for 24 h or 48 h. Then, cells were washed with PBS and stained for 1 h with 1.7 ng/µL Calcein AM viability dye (Ultrapure grade, eBiosciences, San Diego, CA, USA) (24 h viability) or 30 min with Calcein Red AM (Biolegend, San Diego, CA, USA) (48 h viability). Fluorescence activity was determined by flow cytometry analysis using a Guava^®^easyCyteTM (Millipore/Merck, Burlington, NY, USA) according to manufacturer’s instructions.

### 2.4. RNA Isolation and RT-qPCR

RNA was isolated from monocytes and differentiated macrophages using TRIzol^®^ (Invitrogen) and the RNeasy Mini Kit (Qiagen, Hilden, Germany) according to the manufacturer’s instructions. cDNA was generated using the iScript™ cDNA synthesis kit (Bio-Rad, Hercules, CA, USA) and RT-qPCR testing was performed by using SYBRGreen detection chemistry with SsoFast EvaGreen Supermix (Bio-Rad) on a CFX96™ Real-Time PCR Detection System (Bio-Rad). Data were analyzed using the 2^−ΔΔCq^ method with *HPRT1* for normalization [45]. For absolute quantification of mRNA abundance of *ABCD1* and *ABCD2*, standard curves were generated from known copy numbers of linearized plasmids containing *ABCD1*, *ABCD2*, and *HPRT1* cDNA. The sequences of the primers used are listed in Appendix A. 

### 2.5. β-Oxidation of 14C-Labelled C16:0 and C26:0

Degradation rates of the VLCFA C26:0 and the LCFA C16:0 were assessed in primary human macrophages differentiated with GM-CSF and polarized for 48 h with 100 ng/mL LPS (Sigma Aldrich) and 25 ng/mL IFNγ. The effect of the HDACi was assessed in macrophages differentiated with M-CSF and polarized with 100 ng/mL IL-4 (Novartis, Basel, Switzerland) for 48 h in the presence of either DMSO solvent, 2 µM vorinostat, entinostat, or tefinostat. Radiolabelled fatty acids, [1-14C]-palmitic acid (C16:0; ARC 0172A], and [1-14C]-hexacosanoic acid (C26:0; ARC 1253), were obtained from American Radiolabeled Chemicals (St. Louis, MO, USA). Free fatty acids in ethanol were aliquoted into glass reaction tubes, dried under a stream of nitrogen, and solubilized in 10 mg/mL α-cyclodextrin using ultrasonication. The reaction mix contained 4 µM of labelled fatty acids, 2 mg/mL α-cyclodextrin, 30 mM KCl, 8.5 mM ATP, 8.5 mM MgCl2, 1 mM NAD+, 0.17 mM FAD, 2.5 mM L-carnitine, 0.16 mM CoA, 0.5 mM malate, 0.2 mM EDTA, 1 mM DTT, 250 mM sucrose, and 20 mM Tris-Cl, pH 8.0. Reactions were started by addition of 5 × 10^6^–2 × 10^7^ cells, carried out for 1 h at 37 °C and stopped by addition of KOH and heating to 60 °C for 1h. After protein precipitation by HClO4, a Folch partition was carried out, and the amount of released 14C-acetate was determined in the aqueous phase by using a Tri-Carb 4910TR Scintillation Counter (Perkin Elmer, Waltham, MA, USA) and normalized to the protein content of the cell sample. 

### 2.6. Immunoblotting

Differentiated macrophages untreated or polarized with LPS and IFNγ for 48 h were lysed in RIPA buffer containing protease inhibitors (cOmplete, Roche, Basel, Switzerland) and 5× sample buffer. Proteins were separated in a denaturing 7.5% polyacrylamide gel by discontinuous electrophoresis (SDS-PAGE) and semidry blotting into a nitrocellulose membrane. Blots were blocked with 4% non-fat dry milk powder (*w*/*v*) in TBST-T and probed with primary mouse antibodies against the human ABCD1 protein (1:10,000, ALD-1D6-AS, clone 2AL-1D6, Euromedex, Strasbourg, France) or anti-human β-actin (1:100,000, Chemicon, Tokio, Japan). The horseradish peroxidase-conjugated anti-mouse secondary antibody (1:30,000, Dako, Gluostruo, Denmark) was used together with the Immobilon Western HRP Substrate Peroxide Solution and Immobilon Reagent (Millipore) to detect protein expression in a ChemiDoc Imaging System with the Image Lab software 6.0.1(Bio-Rad).

### 2.7. Flow Cytometry Analysis

To determine the purity of the isolated CD14+ monocytes, CD3+ T cells, and CD19+ B cells, cells were blocked with 4% Beriglobin P (Cat.No.I4506, Sigma Aldrich) for 30 min on ice and stained with FITC anti-human CD14 REAfinity antibody (Cat.No.130-110-518, Miltenyi Biotec), FITC anti-human CD3 antibody (Cat.No.130-080-401, Miltenyi Biotec), or PE anti-human CD19 antibody (Cat.No.302207, BioLegend, San Diego, CA, USA) for 30 min (Appendix A). For measurement of protein acetylation levels using flow cytometry analysis, CD14+ monocyte-differentiated macrophages, CD14+ monocytes, CD19+ B cells, and CD3+ T cells were treated with either 2 µM vorinostat, tefinostat, or the DMSO solvent control for 6 h. Collected cells were fixed on ice with 1% paraformaldehyde for 15 min, washed with PBS, and permeabilized with 0.1% Triton X-100 for 10 min at room temperature. Before staining, cells were blocked with 4% Beriglobin P for 30 min on ice and then incubated with 150 ng/mL of the primary antibody anti-acetylated lysine (Ac-K-103 Cat.No.9681, Cell Signaling Technology, Danvers, MA, USA) for 1 h at 4 °C. Upon washing, cells were incubated with the secondary FITC-conjugated anti-mouse antibody (Cat.No.F0479, Dako) at 1:10 dilution for 45 min at 4 °C. For analysis of CD36 protein levels, macrophages were treated with vorinostat, entinostat, tefinostat (2 µM), or the DMSO solvent control for 24 h. Harvested cells were washed and blocked with 4% Beriglobin P for 30 min at 4 °C and then stained with APC-conjugated mouse anti-human CD36 (Cat.No.130-110-741, Miltenyi Biotec) or APC-conjugated mouse anti-human isotype control (Cat.No.130-113-434, Miltenyi Biotec) antibodies. Data were acquired using a Guava^®^easyCyteTM flow cytometer (Merck, Kenilworth, NJ, USA) with the GuavaSoft 3.4 software.

### 2.8. Real-Time Myelin Phagocytosis and Migration Assay

To investigate the phagocytic capacity, M-CSF differentiated macrophages were treated with either vorinostat, entinostat or tefinostat (2 µM) or DMSO solvent control for 24 h before 20 µg/mL of pHrodoTM-green labelled murine myelin was added. The myelin was isolated from brain tissue of 6 to 8-week-old wild-type C57BL6/J mice and stained with pHrodoTM-green dye (Thermo Fischer Scientific, Waltham, MA, USA) as described previously [2]. After the addition of labelled myelin, the phagocytic capacity was assessed by live-cell imaging using the IncuCyte^®^ Live-Cell SX5 analysis system (Sartorius AG, Göttingen, Germany) with fluorescent activity being recorded every 30 min for 20 h. Data were analyzed and extracted using the IncuCyte 2021C software with the area of the green signal normalized to the contrast phase area for each of the images. For the chemotaxis assay, macrophages were pre-incubated with either entinostat, vorinostat or tefinostat (2 µM) or DMSO solvent control for 2 h before cells were pro-inflammatory activated by the addition of 100 ng/mL LPS for 24 h. After treatment, the supernatant was harvested, centrifuged at 5500 RCF at 4 °C for 2 min and concentrated 8-fold using Amicon^®^ 10 KDa MWCO columns (Cat.No.UFC801008, Merck) for use as chemoattractant. The capacity of human primary PBMCs to migrate towards the supernatant of treated vs. untreated macrophages was tested using matrigel-coated membranes (Cat.No.354277, Corning, Corning, NY, USA), Clearview 96-well Plates (Cat.No.4582, Sartorius, Göttingen, Germany) and IncuCyte Live-Cell SX5 imaging. The chemotactic response (cell migration from the top to the lower chamber) was determined as the phase object area (cell covered area) disappearing from the topside of the membrane normalized to the initial value. 

### 2.9. Cytokine Measurement

Differentiated human macrophages were pre-treated with either vorinostat, entinostat, tefinostat (2 µM), or the DMSO solvent control for 2 h prior to pro-inflammatory stimulation with 100 ng/mL LPS (Sigma Aldrich) for 4 h. Then, the supernatant was collected, cell debris was removed by full-speed centrifugation for 2 min at 4 °C, and IFNγ, IL-1β, IL-5, IL-10, IL-12p70, and IL-22 protein levels were determined by Simoa CorPlexTM Human Cytokine 10-plex Panel 1 assay (Cat.No.85-0329, Quanterix, Billerica, MA, USA) on an SP-X imaging and analysis system (Quanterix, Billerica, MA, USA). For GM-CSF levels, a Simoa GM-CSF 2.0 Kit (Cat.No.102329, Quanterix) and a Simoa HD-X Analyzer (Quanterix, Billerica, MA, USA) were used. After diluting the cell supernatant 4-fold (HD-X analysis) or 1:4.75 (SP-X analysis) in sample diluent buffer, assays were performed following the manufacturer’s protocols. The cytokines/chemokines CCL4, CCL7, IL-15, IL-12p40, and CXCL8 were measured using Luminex^®^ technology and the MILLIPLEX MAP Human Cytokine/Chemokine Magnetic Bead Panel (HCYTOMAG-60K-06, Merck) according to the manufacturer’s instruction. 

### 2.10. Bioinformatic Analysis

Transcriptomics data from the rim of chronic active MS lesions (GSE108000, created by Hendrickx et al. [46]) and from human macrophages differentiated in the presence of myelin debris (GSE245235) isolated from brain tissue of 6–8 week-old wild-type C57BL6/J mice, as described previously [41], were reanalyzed for differential gene expression of peroxisome-associated genes using the Qlucore Omics Explorer Software 3.9 (Qlucore AB, Lund, Sweden). Raw counts were normalized using the Trimmed Mean of M-values (TMM) method and the heatmap was created using log-transformed values and scaling with Z-scores. For pathway analysis, the reactome pathway subset provided by the Qlucore Omics Explorer software was used, and the resulting hits were ranked based on the number of matched genes from the total gene set involved in the pathway. 

### 2.11. Statistics

For statistical analysis, two-tailed Student’s *t*-test and a mixed model followed by Dunnett’s or Tukey’s post-hoc test were used. The *p*-values below 0.05 were regarded to indicate statistical significance. Graphs were produced and statistical results were calculated using GraphPad Prism 8. Boxplots indicate median ± interquartile ranges, while whiskers show minimum and maximum ranges. Bar graphs show individual data points with means ± standard deviations (S.D.).

## 3. Results

### 3.1. Peroxisomal Genes Involved in VLCFA Degradation Are Induced by Myelin Phagocytosis but Downregulated in the Brain White Matter of MS Patients

During neuroinflammation, phagocytes clean the damaged area from VLCFA-enriched myelin debris. Thus, phagocytes depend on their peroxisomal β-oxidation activity to metabolize the excess of VLCFAs upon myelin uptake. The degradation of VLCFAs in peroxisomes requires the import of activated fatty acyl-CoAs into the lumen of peroxisomes by the VLCFA transporter ABCD1 or the functionally related ABCD2 protein. Within the peroxisomal matrix, β-oxidation takes place. Acyl-CoA oxidase 1 (ACOX1) catalyzes the first reaction in the degradation of VLCFAs, converting acyl-CoAs to 2-trans-enoyl-CoAs, which are subsequently converted in a two-step process to β-ketoacyl-CoA by the D-bifunctional protein (DBP) encoded by the *HSD17B4* gene. Finally, acetyl-CoA acyltransferase 1 (ACAA1) cleaves the β-ketoacyl-CoA, releasing acetyl-CoA and a fatty acyl-CoA shortened by two carbons (Figure 1A). This process is repeated to generate shorter chain fatty acyl products that can be further metabolized in the mitochondria. Using RNA-sequencing (GSE245235), we investigated how the phagocytosis of myelin debris during the in vitro differentiation of macrophages, mimicking the infiltration of monocytes into active demyelinating lesions, impacts the expression of these genes involved in the breakdown of VLCFAs (Figure 1B). Our analysis revealed that the uptake of myelin significantly stimulated the gene expression of the rate-limiting *ABCD1* and the three enzymes involved in VLCFA β-oxidation (Figure 1B, Appendix A).

We next assessed how neuroinflammation affects peroxisomal gene expression in the context of MS with the prominent involvement of myelin-loaded phagocytes in brain lesions. We retrieved data from a transcriptomic study by Hendrickx et al. that investigated the expression profile of MS pathology in brain white matter lesions [46]. We focused on the rim regions of chronic active MS lesions that highly correlate with disease progression and are characterized by the presence of inflammatory foamy lipid-laden microglia/macrophages resulting from myelin phagocytosis (Figure 1C). By reanalyzing this data set specifically for genes associated with peroxisomes, we observed a prominent and general dysregulation of peroxisomal genes (Figure 1D). A reactome pathway analysis using the Qlucore Omics Explorer Software revealed that the peroxisomal genes associated with β-oxidation and involved in the import and degradation of VLCFAs were among the top candidates being downregulated in chronic active rim regions of MS patients compared to healthy control white matter (Figure 1E). Together, our data show that long-term myelin uptake induces the expression of genes involved in the peroxisomal degradation of VLCFAs in healthy macrophages, while in the rim of chronic active MS lesions, an area enriched in lipid-laden phagocytes, β-oxidation genes involved in VLCFA breakdown are downregulated.

### 3.2. ABCD1 Encoding the Rate-Limiting Factor of VLCFA Degradation and Stimulated by Pro-Inflammatory Activation, Is Downregulated in MS Macrophages

In MS lesions, phagocytes not only adopt a foamy morphology due to the excessive uptake of myelin debris and their disturbed lipid metabolism but are also exposed to a pro-inflammatory environment. Thus, we next evaluated how the exposure to LPS and IFNγ affects VLCFA degradation, and compared the expression of genes involved in peroxisomal β-oxidation between macrophages derived from MS patients and healthy controls. We first isolated human CD14+ monocytes from healthy donors, differentiated them in vitro to macrophages, and polarized them into pro-inflammatory cells using LPS and IFNγ for 48 h. We assessed the peroxisomal β-oxidation rate of the VLCFA C26:0 and, for comparison, the degradation of the saturated long-chain fatty acid C16:0, predominantly degraded in mitochondria, in homeostatic and LPS/IFNγ activated macrophages. For this, we measured the release of water-soluble radioactively labelled acetyl-CoA. Upon 48 h of pro-inflammatory LPS/IFNγ polarization, we observed a significant increase of the peroxisomal β-oxidation rate, whereas the mitochondrial breakdown of C16:0 remained unchanged (Figure 2A). This induction of peroxisomal VLCFA degradation in LPS/IFNγ-treated control macrophages was reflected by the upregulation at the protein level of the rate-limiting ABCD1 and, at the RNA level, by the peroxisomal β-oxidation gene *ACOX1* (Figure 2B,C and Appendix A). Next, we investigated whether the dysregulation of VLCFA metabolism in MS brain tissue would also be reflected in patient-derived macrophages. We used transcriptomics to evaluate monocyte-derived macrophages from patients with MS that were polarized to pro-inflammatory activated cells using LPS/IFNγ. We analyzed this data set for the expression of peroxisomal genes involved in VLCFA degradation (Figure 2D, Appendix A). Our results revealed a significant downregulation of *ABCD1* in MS macrophages before and after stimulation with LPS/IFNγ when compared with healthy control macrophages. In addition to the rate-limiting factor *ABCD1*, other genes involved in peroxisomal VLCFA degradation were also differentially expressed in MS macrophages (Appendix A). Of note, macrophages from MS patients did not significantly upregulate the expression of *ABCD1* upon pro-inflammatory stimulation (Appendix A). Together, our data show that healthy control macrophages respond with elevated import and degradation of VLCFAs in peroxisomes during pro-inflammatory conditions, whereas macrophages derived from MS patients show lower expression rates of *ABCD1*, regardless of pro-inflammatory polarization.

### 3.3. Tefinostat Modulates VLCFA Metabolism in Macrophages but It Is Less Effective than Entinostat at Inducing Peroxisomal VLCFA Transporter Expression and Degradation of VLCFAs

We previously found that the loss of ABCD1 function in macrophages derived from patients suffering from X-ALD impairs peroxisomal VLCFA degradation and prolongs pro-inflammatory gene expression upon activation [2,11]. Both the pan-HDACi vorinostat and the class I specific HDACi entinostat induce peroxisomal VLCFA β-oxidation in X-ALD and control macrophages [41,47]. The clinical utility of both vorinostat and entinostat for X-ALD and other neuroinflammatory disorders is limited by its toxicity profile, which requires an intermittent dosage regime [48,49]. We reasoned that selectively targeting an HDACi to macrophages, directly modulating VLCFA metabolism in this disease-relevant cell type, could circumvent side effects.

Tefinostat is a pan-HDACi that is cleaved by the intracellular enzyme human carboxylesterase-1 (CES1), which is enriched in monocytes/macrophages (Figure 3A). Upon cleavage, the acidic form of tefinostat is retained in the cell and accumulates, resulting in increased HDACi activity within cells of the monocytic lineage when compared to other immune cell types such as B and T cells showing low CES1 expression [50]. To confirm the cell-type selectivity of tefinostat, we treated monocytes, monocyte-differentiated macrophages, B and T cells with either tefinostat or vorinostat, a compound structurally similar to tefinostat but lacking cell-type selectivity (Figure 3A), for 6 h before the rate of histone protein acetylation was determined by a flow cytometry analysis. We evaluated the effect of tefinostat at 2 µM, corresponding to a clinically relevant dosage of approximately 320 mg/day, which was based on the clinical study performed by Ossenkoppele and colleagues [43]. Our results revealed significantly increased histone acetylation in the macrophages treated with tefinostat when compared to vorinostat (Figure 3B). However, in contrast to elevated responses in the macrophages, we were unable to confirm the increased efficacy of tefinostat in the monocytes compared to vorinostat (Figure 3B). The viability of the macrophages was not compromised by 24 h and 48 h of HDACi treatment, including entinostat, vorinostat, and tefinostat (Appendix A). 

To investigate whether tefinostat, like entinostat and vorinostat, enhances the catabolism of VLCFAs, we incubated macrophages derived from healthy donors with tefinostat, vorinostat, or entinostat for 24 h and analyzed the expression of the peroxisomal VLCFA transporters and β-oxidation enzymes (Figure 3C). Our data revealed that all three HDACis were able to induce the expression of peroxisomal genes but to a different extent. Whereas only entinostat significantly upregulated the expression of *ABCD1*, all three HDACis stimulated the expression of the *ABCD1* homologue, *ABCD2*, as well as the β-oxidation gene *ACAA1*. Interestingly, *ACOX1* expression was not modulated by any of the HDACi treatments, and *HSD17B4* expression was increased by entinostat and vorinostat but not by tefinostat. Next, we assessed how tefinostat treatment impacts the rate of the peroxisomal and mitochondrial β-oxidation of macrophages upon 48 h treatment. We chose this time point for the translation of the epigenetic effects of the HDACi to the protein level. Although tefinostat induced the degradation of the VLCFA C26:0 to a similar level as vorinostat when compared to the DMSO solvent-treated macrophages, entinostat showed the highest efficacy (Figure 3D). Collectively, these data reveal that although the macrophage-selective, pan-HDACi activity of tefinostat impacts VLCFA metabolism, it is less potent than the class-I selective HDACi entinostat. 

### 3.4. Tefinostat Interferes with the Chemotactic Recruitment of PBMCs

At the onset of neuroinflammation, CNS lesions of patients with MS and X-ALD are characterized by an immune cell breach across the glia limitans barrier. These cells, mostly monocyte-derived macrophages, degrade extracellular matrix molecules to infiltrate the deep CNS parenchyma. Upon migration, they encounter a pro-inflammatory environment and participate in the secretion of cytokines to reinforce further recruitment. Using Boyden chamber endpoint assays, we previously found that entinostat treatment decreased the capability of LPS-activated macrophages to recruit immune cells [41]. Therefore, we next analyzed the effect of tefinostat treatment on immune cell recruitment by LPS-stimulated macrophages using live imaging technology. We first evaluated whether pro-inflammatory conditions would alter the expression of *CES1*, the enzyme responsible for conferring cell-type selectivity to tefinostat, in human macrophages. We found that the pro-inflammatory LPS treatment of macrophages further induced *CES1* expression, which was statistically significant after 48 h (Figure 4A). We next assessed if HDACi treatment would interfere with the ability of macrophages to secrete chemokines and recruit PBMCs upon pro-inflammatory LPS-stimulation. We pre-treated macrophages derived from healthy donors with either entinostat, vorinostat, tefinostat, or the solvent control for 2 h and stimulated them with LPS for an additional 24 h in presence of the inhibitors or the solvent. The supernatants from the treated macrophages were used to assess the recruitment of PBMCs across matrigel (enriched in extracellular matrix molecules)-coated membranes with an IncuCyte live imaging system (Figure 4B). Despite all three inhibitors interfering with PBMC recruitment, only the supernatant from the tefinostat-treated macrophages showed a statistically significant reduction of the PBMC infiltration rate after 12 h (Figure 4C,D). It is noteworthy that when the secreted cytokines were measured in the supernatant of the macrophages as early as 4 h upon pro-inflammatory activation, entinostat was more potent than vorinostat and tefinostat at reducing the secretion of CXCL8, IFNγ, or IL5 (Appendix A). 

### 3.5. Entinostat Has the Highest Capacity to Promote Uptake of Myelin Debris and Expression of Proteins Involved in Lipid Export

Upon phagocytosis, macrophages break down the myelin debris, thereby producing lipid and cholesterol metabolites that bind to and activate the nuclear receptor LXR. LXR is a key transcription factor in modulating inflammation and aiding the reverse cholesterol transport process in macrophages, thus preventing cholesterol build-up and the associated crystal formation (Figure 5A) [51]. To establish the efficacy of tefinostat to stimulate the uptake of myelin, we treated healthy control macrophages with either tefinostat, entinostat, vorinostat, or the DMSO solvent for 24 h and assessed the gene expression of the phagocytic receptors *CD36*, *MER proto-oncogene tyrosine kinase* (*MERTK*), and *macrophage scavenger receptor 1* (*MSR1*). We observed that *CD36*, encoding a receptor involved in myelin phagocytosis and the uptake of saturated LCFA and VLCFA, was significantly upregulated by both entinostat and vorinostat but remained unchanged by tefinostat treatment. In contrast, only entinostat significantly induced the expression of *MERTK* (Figure 5B, CD36 upregulation on the protein level is shown in Appendix A). The variability of cells derived from different donors obscured any potential differences in *MSR1* expression upon the HDACi treatment (Figure 5B). 

Tefinostat cell-type selectivity relies on CES1 enrichment in macrophages; thus, we aimed to evaluate how long-term myelin exposure, as it occurs during chronic neuroinflammatory diseases, affects CES1 expression levels. Using our RNA-seq transcriptomic data from the macrophages differentiated in the presence of myelin, we observed increased *CES1* expression levels upon myelin loading (Figure 5C). Next, we studied whether changes in the expression of the phagocytic receptors *CD36* and *MERTK* upon HDACi treatment were reflected in an increased uptake of myelin debris. To this end, murine myelin debris was fluorescently labelled with the pH-sensitive pHrodo-dye and added to the macrophages that had been pre-treated with entinostat, vorinostat, tefinostat, or the solvent control. The uptake of the pHrodo-labeled myelin by the macrophages was analyzed using an IncuCyte live imaging system (Figure 5D). The results revealed that all three HDAC inhibitors boosted myelin uptake in the early phase of the phagocytic response. This included tefinostat, which, interestingly, did not significantly change the mRNA levels of the phagocytic receptors *CD36*, *MERTK*, or *MSR1* upon 24 h treatment. The response to tefinostat then tapered off more quickly, while entinostat and vorinostat maintained the induction 9 h after myelin addition; again, entinostat had the strongest effect (Figure 5E).

We finally compared the ability of the different HDACis to prevent foam cell formation by upregulating the genes involved in lipid export. While all three HDACis induced the expression of the *ATP-binding cassette subfamily G member 1* (*ABCG1*), entinostat and vorinostat, but not tefinostat, were able to stimulate the expression of *ATP-binding cassette subfamily A member 1* (*ABCA1*). Further, only entinostat induced the expression of *apolipoprotein E* (*APOE*), a protein mediating the delivery of cholesterol and other lipids (Figure 5E). 

## 4. Discussion

Saturated fatty acids are believed to establish a conducive environment within the plasma membrane that facilitates the formation of cholesterol-dependent networks responsible for pro-inflammatory signaling [52]. Primary peroxisomal deficiency, as found in X-ALD, but also the secondary modulation of peroxisomal functions, as previously described for progressive MS [19,20], are associated with an impaired VLCFA metabolism that possibly drives neuroinflammation and neurodegeneration. Recent studies suggest that the inhibition of the synthesis of fatty acids, including VLCFAs, promotes remyelination by inducing a reparative phenotype in phagocytes [53]. In this work, we focused on macrophages (observed in chronic active MS lesions, and active CALD lesions) as one of the main drivers of inflammatory processes in X-ALD or MS and investigated the potential of macrophage-targeted HDAC inhibition to promote peroxisomal VLCFA degradation and the phagocytosis of myelin debris and to interfere with immune cell recruitment. 

The alteration of the cellular lipid composition is a prerequisite for macrophages to modulate their immune function in response to environmental cues. In X-ALD patients, impaired lipid metabolic pathways involving VLCFAs result in pathological changes in the brain. Next to inherited peroxisomal defects, more generally the onset of neuroinflammation, which is accompanied by the destruction of the lipid-enriched myelin sheaths, may result in a dysregulated peroxisomal metabolism, including VLCFA degradation. In attempts to further link peroxisomal VLCFA β-oxidation with the resolution of inflammation and repair, we show that long-term incubation (during differentiation) with myelin in healthy macrophages results in the increased expression of peroxisomal genes involved in β-oxidation, possibly as a response to an excess of myelin-derived VLCFA in the cells. Similarly, prolonged pro-inflammatory stimulation (48 h) with LPS and IFNγ induced the β-oxidation activity for VLCFAs in healthy macrophages. These findings are in line with our previous work showing that the upregulation of peroxisomal VLCFA degradation upon the pro-inflammatory LPS activation of macrophages is required for the resolution of inflammation [11]. To obtain further support for this concept, we re-analyzed transcriptomic data (GSE108000) from white matter tissue of unrelated-disease controls and MS cases, focusing on chronic active lesions, in which the rim is enriched in lipid-laden phagocytes [46]. We found profound changes in peroxisomal gene expression that primarily reflected alterations in the VLCFA β-oxidation pathway. Our results confirm a recent study by Roczkowsky and colleagues, reporting reduced expression of the β-oxidation gene *HSD17B4* and other peroxisomal genes in normal-appearing white matter from MS patients vs. other disease controls [20]. Importantly, the transcriptomic analysis corroborates a repressed expression of *ABCD1*, the rate-limiting factor in peroxisomal VLCFA β-oxidation, in macrophages in MS patients. Furthermore, we can show the downregulation of *ABCD1*, independent of pro-inflammatory stimulation, in ex vivo peripheral monocyte-derived macrophages in MS patients vs. healthy controls. Therefore, peroxisomal alterations appear to be present in MS macrophages even prior to the uptake of myelin debris and formation of lipid-laden pro-inflammatory foam cells. Several recent publications, based on profiling of foamy macrophages/microglial cells in chronic active MS lesions, reveal iron as a possible culprit for a skewing towards a pro-inflammatory phenotype [31,54,55]. In this context, chronic iron overload has previously been linked to lipid peroxidation, damage of the peroxisomal membrane, and the loss of peroxisomal functions, as well as ferroptosis (cell death caused by iron-induced lipid peroxidation [56,57]. However, whether pathological iron accumulation or other intrinsic factors render macrophages sensitive to secondary peroxisomal changes in progressive MS and how these changes impact the MS pathomechanisms remain to be elucidated. 

We have previously described the beneficial effects of HDACi treatment for neuroinflammatory disorders [40,41]. Here, we investigated the potential of macrophage-targeted HDAC inhibition to promote peroxisomal VLCFA degradation and the phagocytosis of myelin debris and to reduce immune cell recruitment. We focused on the use of foamy and pro-inflammatory macrophages, as they are among the main drivers of inflammatory processes in X-ALD and MS. We directly targeted macrophages by using the ESM-conjugated HDACi tefinostat. Tefinostat has been reported to be selectively hydrolyzed and retained within monocytes/macrophages due to their high expression of *CES1*, the enzyme responsible for its cleavage [42,43]. We hypothesized that tefinostat would beneficially modulate peroxisomal lipid homeostasis and inflammatory processes selectively within this disease-relevant cell lineage, while concurrently limiting adverse reactions associated with the use of broad spectrum HDACis. Indeed, despite the high expression of CES1 in human hepatocytes [50], clinical studies have demonstrated the absence of liver-related toxicity of tefinostat and its safety in human patients [43,58]. When we compared the activity of tefinostat to that of vorinostat in various primary human immune cell types, as expected, we observed no effect on protein acetylation in B cells and T cells and a robust increase in monocyte-differentiated macrophages. Surprisingly, however, in monocytes, no significant increase in acetylation could be detected—at least not upon 6 h incubation. This finding contrasts with a previous observation by Ossenkoppele and colleagues who performed flow cytometric pharmacodynamic assays demonstrating increased monocytic protein acetylation in human blood samples from tefinostat-treated patients with hematological malignancies [43]. The divergent outcomes could stem from methodological differences. While our experimental approach encompassed MACS-bead isolation and the in vitro culture of healthy control monocytes, Ossenkoppele et al. examined blood samples directly by using flow cytometric gating strategies. Since that study noted significant protein acetylation specific to monocytes at doses of 320 mg/day, which corresponds to approximately 2 µM in vitro tefinostat according to their pharmacokinetic findings [43], it is unlikely that diminished monocyte specificity in our investigations could be attributed to excessively high or low doses. 

In differentiated macrophages, all three HDACis were able to significantly enhance the degradation of VLCFAs, through the stimulation of peroxisomal β-oxidation and the induction of peroxisomal gene expression, and to promote myelin phagocytosis and the expression of lipid transporters. Nonetheless, a more robust effect was obtained with entinostat, with the exception of preventing the macrophage-induced chemotactic recruitment of PBMCs, wherein tefinostat was more effective, while entinostat or vorinostat-treated macrophages elicited insignificant changes. It is worth noticing that tefinostat selectivity depends on the expression of CES1. In this work, we show that *CES1* expression is induced under prolonged myelin loading and pro-inflammatory stimulation. Thus, we hypothesize that in neuroinflammatory diseases such as MS and X-ALD, CES1 levels might be increased in pro-inflammatory lipid-laden phagocytes that accumulate in the lesion area, possibly further augmenting the selectivity of tefinostat for pro-inflammatory foamy macrophages.

HDACs form a large family of enzymes in humans, encompassing 18 different isoforms that are grouped into three classes (I, II, and IV). In contrast to tefinostat and vorinostat, which act as pan-HDACi, the second-generation classified HDACi entinostat selectively targets class I HDACs [59,60]. Class I HDACs show high expression levels in brain tissues and are implicated in various neural functions [61], suggesting that class I HDACs may be pivotal for the beneficial modulation of peroxisomal lipid metabolism as well as pro-inflammatory functions in human brain cells including macrophages. It is noteworthy that the isoform-specific inhibition of HDAC3 or HDAC8, both of which are class I HDACs with similar structural and neurotoxic properties, was recently reported to ameliorate neuroinflammation and promote neuroprotection by the suppression of glial activation in two different neuroinflammatory mouse models [62,63].

Together, our work demonstrates the high potential for HDACis in the modulation of lipid metabolism and, thereby, shifting macrophages towards a regenerative phenotype in neuroinflammatory diseases. We observed that the pan-HDACi tefinostat, while showing selective activity in macrophages, was less potent than the class I HDACi entinostat in stimulating peroxisomal VLCFA metabolism and innate immune cell functions towards promoting a regenerative macrophage phenotype. By highlighting the intricate relationship between peroxisomal lipid metabolism and other macrophage functions linked to neuroinflammation, our study emphasizes the importance of targeting specific HDAC isoforms to achieve optimal therapeutic outcomes. Further investigation into the precise isoforms of class I HDACs and their downstream targets involved in peroxisomal lipid metabolism in the context of neuroinflammation and demyelinating diseases are required. In addition, the development of class I HDAC-specific entinostat derivatives with selectivity for macrophages could pave the way for more targeted interventions and thus, improve clinical outcomes. 

## Figures and Tables

**Figure 1 biomolecules-13-01696-f001:**
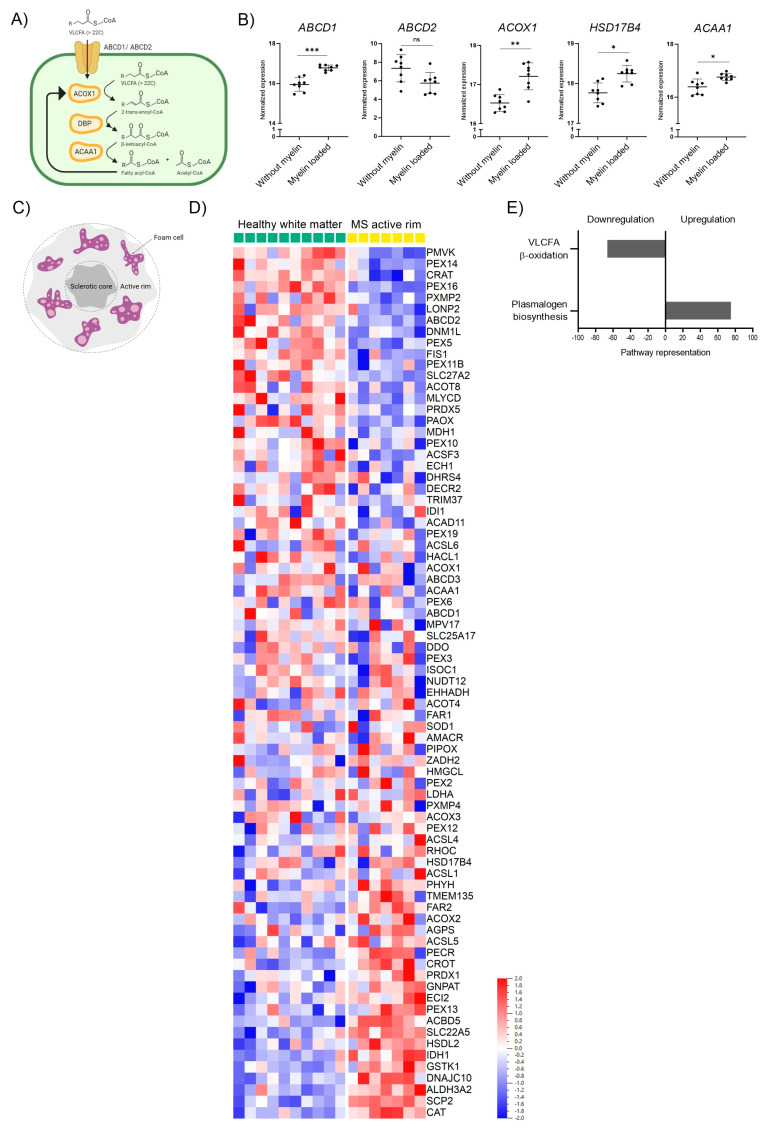
Peroxisomal gene expression is altered upon myelin phagocytosis in human macrophages and in rim regions of MS chronic active brain white matter lesions. (**A**) Scheme indicating the enzymes involved in peroxisomal β-oxidation (ATP-binding cassette transporter D1, ABCD1; ATP-binding cassette transporter D2, ABCD2; acyl-coenzyme A oxidase 1, ACOX1; D-bifunctional protein, DBP; acetyl-CoA acyltransferase 1, ACAA1; encoded by the genes *ABCD1*, *ABCD2*, *ACOX1*, *HSD17B4*, and *AACA1*, respectively. (**B**) RNA-seq transcriptional profiling of genes involved in peroxisomal degradation of saturated VLCFAs in human healthy control macrophages (*n* = 8) differentiated in the presence of murine myelin debris (GEO: GSE245235). Other peroxisomal genes are shown in Appendix A. (**C**) Schematic representation of the active rim in the MS white matter lesion. (**D**) mRNA levels from the active rim lesion of 7 MS patients with relapse-remitting (*n* = 6) or secondary progressive (*n* = 1) MS, and healthy white matter from 10 control individuals (GEO: GSE108000), normalized by Hendrickx et al., 2017 [46] and re-analyzed using the Qlucore Omics Explorer Software (Qlucore AB). The heatmap was created using log-transformation of values and scaling with Z-scores. (**E**) Altered peroxisomal metabolic pathways in MS active rim brain white matter lesions vs. healthy brain white matter. The pathway analysis was performed using the reactome pathway subset provided by Qlucore Omics Explorer Software, excluding pathways conformed by only one gene and a minimum of two genes matched. The number of matched genes per pathway was divided by the gene size of said pathway and this value was expressed as a percentage to depict the level representation of each of the pathways. Only the most upregulated and downregulated pathways were depicted. For statistical analysis, two-tailed paired Student’s *t*-test was used (*** *p* ≤ 0.001, ** *p* ≤ 0.01, * *p* ≤ 0.05; ns = not significant). The graphs in (**B**) indicate means ± S.D. Illustrations in (**A**,**C**) were created with BioRender.com, accessed on 5 September 2023.

**Figure 2 biomolecules-13-01696-f002:**
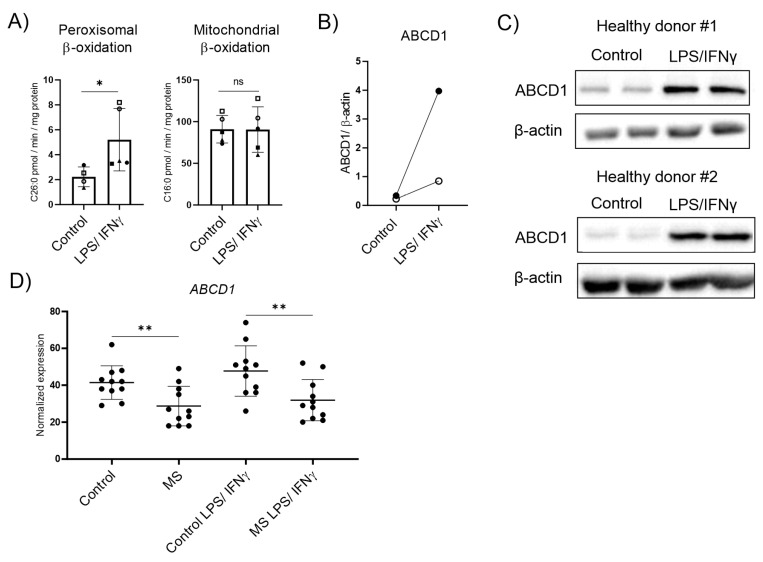
Peroxisomal VLCFA-degradation and expression of the rate-limiting ABCD1 upon pro-inflammatory LPS/IFNγ polarization in healthy control and MS macrophages. (**A**) Mean values of C26:0 and C16:0 degradation by peroxisomal and mitochondrial β-oxidation, respectively, normalized to protein content in primary human LPS/IFNγ polarized macrophages (*n* = 5). (**B**) Protein levels of ABCD1 in human monocyte-derived macrophages from healthy donors untreated or polarized for 2 days with LPS and IFNγ, analyzed by western blot analysis and normalized to β-actin (*n* = 2). Open dots correspond to donor #1 and filled dots correspond to donor #2. (**C**) Immunoblots corresponding to (**B**) showing ABCD1 and β-actin protein expression in untreated and polarized human monocyte-derived macrophages derived from two healthy donors with two technical replicates each. (**D**) RNA-seq transcriptional data of *ABCD1* in healthy control and MS macrophages untreated or polarized with LPS/IFNγ (*n* = 11). The expression of other peroxisomal genes is shown in Appendix A. For statistical analysis, two-tailed paired (**A**) and unpaired (**D**) Student’s *t*-test were used (** *p* ≤ 0.01, * *p* ≤ 0.05; ns = not significant). The graphs in (**A**,**D**) display all data points and means ± S.D. Uncropped images of (**C**) can be found in Appendix A. Symbols were used to visualize data derived from same donors.

**Figure 3 biomolecules-13-01696-f003:**
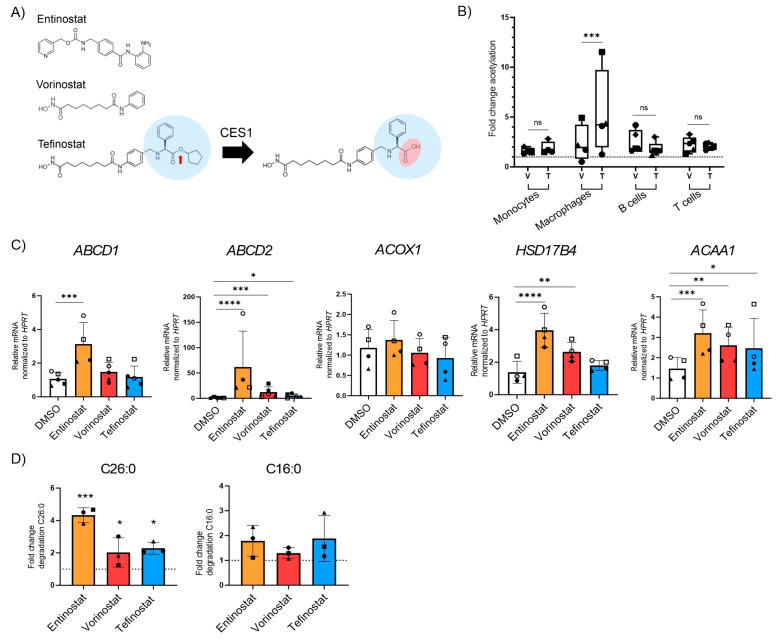
The class one specific HDAC inhibitor entinostat most potently stimulates peroxisomal VLCFA degradation and gene expression in human primary macrophages. (**A**) Scheme indicating the cleavage of tefinostat by the intracellular human carboxylesterase-1 (CES1), resulting in the negatively charged acidic form of tefinostat that is retained within the cell. (**B**) Histone protein acetylation determined by FACS in human monocytes (*n* = 4), monocyte-derived macrophages (*n* = 4), B cells (*n* = 5), and T cells (*n* = 5) isolated from the blood of healthy donors and treated with either vorinostat (V), tefinostat (T, 2 µM), or DMSO as solvent control for 6 h. Data is shown as fold increase relative to DMSO treated samples and depicted as boxplots. (**C**) RT-qPCR testing for *ABCD1*, *ABCD2*, *ACOX1*, *HSD17B4*, and *AACA1* upon treatment of primary human macrophages with entinostat, vorinostat, or tefinostat (2 µM) for 24 h (*n* = 4–5). Data were normalized to HPRT1. (**D**) Degradation rates of C26:0 (**left**) and C16:0 (**right**) by peroxisomal and mitochondrial β-oxidation were determined in primary human macrophages (*n* = 3) treated with either entinostat, vorinostat, tefinostat (2 µM), or DMSO for 48 h. Data are shown as fold increase relative to DMSO treated samples. To make interindividual variants more comparable for the statistical analysis, the qRT-PCR and β-oxidation raw data was log transformed. Two-tailed paired Student’s *t*-test (**B**) and a mixed model followed by Dunnett’s (**C**) or Tukey’s post-hoc (**D**) were used for statistical analysis (**** *p* ≤ 0.0001, *** *p* ≤ 0.001, ** *p* ≤ 0.01, * *p* ≤ 0.05; ns = not significant). Bar graph in (**A**) shows median ± interquartile range while graphs in (**C**,**D**) indicate means ± S.D. The dashed lines indicate normalization to DMSO solvent control. Symbols were used to visualize data derived from same donors.

**Figure 4 biomolecules-13-01696-f004:**
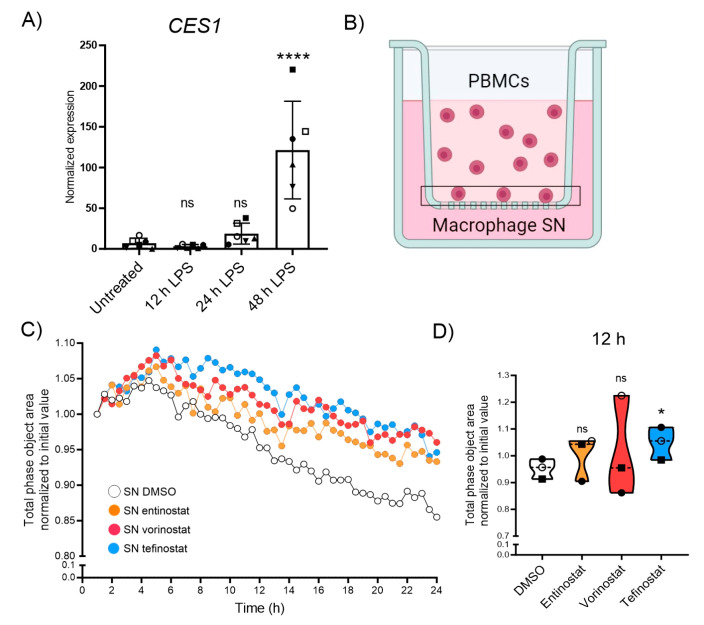
Migration of PBMCs towards macrophage-derived supernatant is antagonized by HDACi treatment. (**A**) Expression levels of *CES1* measured using RT-qPCR testing from human monocyte-derived macrophages upon 0, 12, 24, and 48h of pro-inflammatory stimulation with LPS (*n* = 6). (**B**) Experimental setup of the IncuCyte real-time migration assay towards supernatant derived from human primary macrophages pre-treated with either entinostat, vorinostat, tefinostat (2 µM), or DMSO solvent control for 2 h followed by 24 h LPS-treatment to determine prevention of leukocyte recruitment, and migration across matrigel-coated membrane by HDACi. Vertical-directed cell migration towards the macrophage supernatant (applied to the reservoir) was quantified by the decrease of cell total phase area on the top of the insert membrane over time. (**C**) Mean migration of PBMCs towards supernatant derived from macrophages, pre-treated with entinostat, vorinostat, tefinostat, or DMSO, and then LPS activated over a period of 24 h, with pictures taken every 30 min (*n* = 3). (**D**) PBMC infiltration, measured as the reduction of cell area from the topside of the insert normalized to the initial area, 12 h after application of the chemoattractive, macrophage-derived supernatant (*n* = 3). For statistical analysis, a mixed model with Greenhouse–Geisser correction and Dunnett’s post hoc test was used (**** *p* ≤ 0.0001, * *p* ≤ 0.05; ns = not significant). Bar graph in (**A**) shows means ± S.D. while graph in (**D**) indicates median ± interquartile range. Symbols were used to visualize data derived from same donors. Illustration in (**B**) was created with BioRender.com, accessed on 5 September 2023.

**Figure 5 biomolecules-13-01696-f005:**
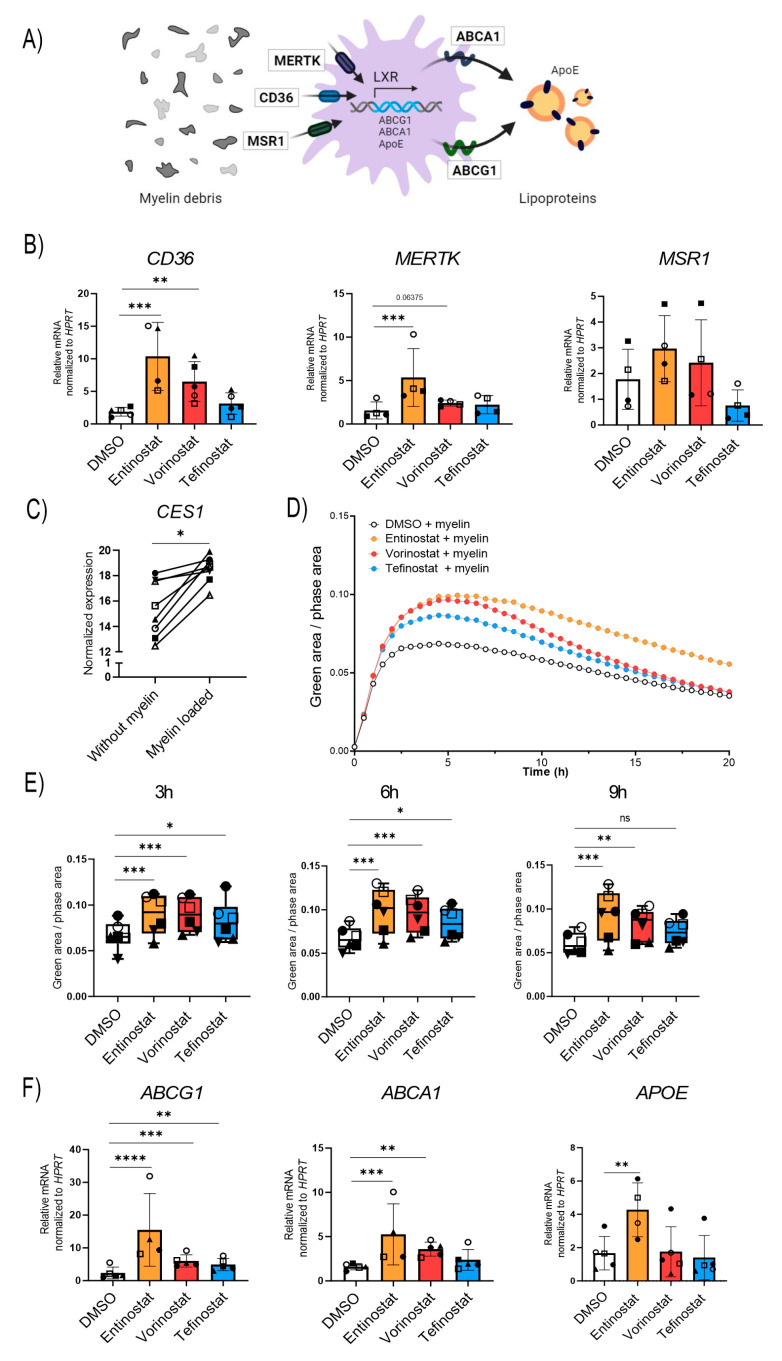
Entinostat most effectively stimulates phagocytosis of myelin debris and upregulates expression of genes involved in lipid export. (**A**) Scheme showing the receptors involved in the uptake of myelin debris (*CD36*, *MSR*-*1*, and *MERTK*), the proteins responsible for the export of lipids (*ABCA1*, *ABCG1*, and *APOE*) and the transcription factor, liver X receptor (LXR), responsible for their synthesis. (**B**) RT-qPCR testing of the phagocytic receptors *CD36*, *MSR-1*, and *MERTK* in human primary macrophages derived from healthy donors pre-treated with entinostat, vorinostat, tefinostat (2 µM), or the solvent control DMSO for 24 h (*n* = 4–5). Data were normalized to *HPRT1*. (**C**) RNA-seq transcriptional levels of *CES1* in human healthy control macrophages (*n* = 8) differentiated in the presence of murine myelin debris (GSE245235). (**D**) IncuCyte live imaging of myelin debris phagocytosis in HDACi or solvent-treated human primary macrophages (*n* = 6). Fluorescence intensity was measured in the IncuCyte imaging platform with 30 min intervals for up to 20 h and (**E**) the phagocytic capacity was compared between entinostat, vorinostat, tefinostat, or DMSO solvent-treated macrophages 3 h, 6 h, and 9 h after myelin debris uptake (*n* = 6). (**F**) RT-qPCR tests of the lipid transporters *ABCG1, ABCA1*, and *APOE* in human primary macrophages derived from healthy donors pre-treated with entinostat, vorinostat, tefinostat (2 µM), or the DMSO solvent control for 24 h (*n* = 4–5). To make interindividual variants more comparable for the statistical analysis, the RT-qPCR data was log transformed. A mixed model followed by Dunnett’s post-hoc test was used for statistical analysis (**** *p* ≤ 0.0001, *** *p* ≤ 0.001, ** *p* ≤ 0.01, * *p* ≤ 0.05; ns = not significant). Bar graphs in (**B**,**E**,**F**) indicate means ± S.D. Symbols were used to visualize data derived from same donors. Illustration in (**A**) was created with BioRender.com, accessed on 5 September 2023.

## Data Availability

All data presented in the manuscript and/or Appendix A are available upon request. RNAseq data corresponding to human monocyte-derived macrophages differentiated in vitro in the presence or absence of myelin from brain tissue of wild-type C57BL6/J mice can be found under the GEO SuperSeries GSE245235.

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
