# Peer review of "Efficacy of HDAC Inhibitors in Driving Peroxisomal β-Oxidation and Immune Responses in Human Macrophages: Implications for Neuroinflammatory Disorders"

_biomolecules, 2023, doi:10.3390/biom13121696_

Round 1

Reviewer 1 Report

Comments and Suggestions for Authors

Overall, the paper is well written with minor issues that can easily be resolved. It clearly introduces the problematic and present their hypothesis. The methods and results are adequately presented and support their conclusions.

Minor corrections:

Line 204 – space between 2.5 and β-oxidation is missing.

Line 207 – Correct INFϒ to IFNϒ.

Line 340 – Refer at the end of the sentence the scheme Figure 1C. In fact, Figure 1C should be divided into two panels to be correctly introduced in the text. (Figure 1C – schematic representation; Figure 1D - heatmap result; Figure 1E – altered peroxisomal metabolic pathways).

Line 392 – To be coherent “Suppl.” should be Supplementary (review this in the rest of the document)

Line 438 – In Figure 2B the authors show an immunoblot of ABCD1 and then refer to Supplementary Figure 3 where they show the same result with a different immunoblot, besides ABCD1 blot quantification. The authors should remove this supplementary figure and include ABCD1 protein quantification in Figure 2. Moreover, in the legend of Figure 2 the authors should indicate that the immunoblot is representative of two biological replicates (n=2), as indicated in the supplementary figure.

Line 445 – The authors indicate Supplementary Fig. 4 but should be corrected to Supplementary Figure 5. They also should indicate the number of biological replicas (n=?). Review all legends in the manuscript since this information is missing in others as well.

Line 510 – In Figure 3D legend the authors indicate that the duration of treatment with the inhibitors was 48h, however previously they used 24h.  They should indicate why did they increased the time of incubation. In line with this, Supplementary Figure 6, which shows macrophage viability upon treatment with each inhibitor should include the 48h time point.

Line 522 – In the previous experiences the authors have always used LPS and IFNϒ to induce macrophages activation and in here they chose to use LPS alone. This introduces a variability since LPS alone is less potent to induce cytokine production. This should be clarified.

Line 565 – Indicate the duration of treatment since previously different incubation times were used.

Line 584 – The phrase should be rewritten to be more clear - The results revealed that all three HDAC inhibitors boosted myelin uptake in the early phage of phagocytic response. Interestingly, even tefinostat treatment boosted myelin uptake when it did not change mRNA levels of phagocytic receptors CD36, MERTK or MSR1.  

Line 677 – Reference is missing

Comments on the Quality of English Language

In general, the quality of English languase is good however, sometimes the authors use long phrases which may be hard to follow.

Reviewer 2 Report

Comments and Suggestions for Authors

In general, this article demonstrates an excellent level of quality in various aspects. The language used throughout the manuscript is clear and professional, contributing to the overall readability and comprehension of the content. The methods employed in the research are robust and well-documented, ensuring the reliability of the results obtained. Speaking of results, they are presented coherently and discussed comprehensively, which significantly enhances the understanding of the study's findings.

Given the high quality of language, methodology, results, and discussion, I would recommend that this article be considered for publication, after minor revisions:

The provided title, while descriptive, could benefit from more clarity and specificity. A more suitable title for the research article could be: "Assessing Tefinostat's Impact on VLCFA Metabolism and Immune Responses in Human Macrophages: Implications for Neuroinflammatory Disorders", by including specific variables in your study.

The introduction section of this article appears to be excessively detailed, encompassing information that could be more effectively summarized. For example, details related to clinical trials might be best condensed within the introduction, reserving in-depth analysis for the discussion section, as highlighted in references 41-43. Streamlining the introduction would enable a more concise and impactful presentation of the core research objectives.

Reviewer 3 Report

Comments and Suggestions for Authors

In this manuscript, diverse characteristics of peroxisomal beta-oxidation in wild type and MS macrophages were studied. Furthermore, the efficacy of the macrophage selective HDAC inhibitor tefinostat was assessed on peroxisomal beta-oxidation and on the macrophage phenotype. The paper is well written with an extensive introduction and data are nicely presented.

In the abstract nothing is mentioned about the two first of the five parts of the results section.

Line 341-350: there is dysregulation of peroxisomal genes in the MS rim region according to the re-analysis of transcriptome data. Only the downregulation of some genes involved in peroxisomal beta-oxidation is discussed and termed ‘differently’ expressed. It would be better to define this as ‘downregulated’. In the figure, there seems to be large variability in expression patterns among peroxisomal genes. In fact the crucial ABCD1 and ACOX1 genes of peroxisomal beta-oxidation are not clearly differentially expressed in MS tissue. With regard to PEX genes, some are up- and others are downregulated. This raises the question what the mechanism might be underlying these transcriptional changes. Based on these data it does not seem justified to conclude that “β-oxidation genes involved in VLCFA breakdown are downregulated”.

In Figure 2C, the effect of LPS/IFN on ABCD1 transcripts in WT cells is very limited and apparently not significant. This does not seem to be so different from the LPS/IFN effect on MS . The former is in contrast to the induction of ABCD1 protein levels (Figure 2B). How can all these data be reconciliated?

Minor comments

Line 50: there are 3 subtypes of PPAR with diverse expression and function. Which type is referred to here?

Line 60: second part of the sentence only refers to VLCFA. It should be made clear that peroxisomes are not involved in cholesterol breakdown

Line 125: fibroblasts

Line 204: typo in title

Line 435: rated should be rates?

Please check spelling of the HDAC inhibitors throughout the manuscript
